# Health impact of the COVID-19 in Spanish non-healthcare workers by gender: Use of sickness absence for surveillance

**Dante R. Culqui L.**[1], **Alín Manuel Gherasím**[2]*, **Sofía Escalona López**[1], **Jesús Oliva Domínguez**[2], **Montserrat García Gómez**[2]*

1 External Senior Technician (TRAGSATEC), Occupational Health Unit, Subdirectorate General for Environmental Health and Occupational Health, General Directorate for Public Health and Health Equity, Spanish Ministry of Health, Madrid, Spain, 2 Occupational Health Unit, Subdirectorate General for Environmental Health and Occupational Health, General Directorate of Public Health and Health Equity, Spanish Ministry of Health, Madrid, Spain

* mgarciag@sanidad.gob.es (MGG); agherasim@sanidad.gob.es (AMG)

**Data Availability Statement:** The data underlying the results presented in the study are available,

## Abstract

### Introduction

At the beginning of the pandemic, the identification of transmission chains was biased towards more closely monitored sectors, such as healthcare and sociosanitary centers.

### Objective

The objective of our research is to describe the impact of the pandemic on the Spanish non-healthcare workers measured through health-related workplace absences.

### Methods

A descriptive study of the cases of COVID-19-related temporary disability (TD) between February 15th and September 17th, 2020, was carried out. TD quarantine/infection ratios were calculated for every economic sector of activity. Rates of COVID-19 TDs per 100,000 affiliated workers were obtained, by gender, age, economic activity of the company or occupation of the worker.

### Results

A total of 1,126,755 TDs were recorded, 45.4% in women. The overall TD rates were 5,465 quarantines and 1,878 illnesses per 100,000 women, and 4,883 quarantines and 1,690 illnesses per 100,000 men. The highest incidence rates of TD due to illness were observed in younger age groups, under 30. The median TD quarantine/infection ratio was 2.6 (Interquartile range [IQR] 1.5–3.9), and *Postal and Courier activities* had the highest value: 4.7 quarantines per case (IQR: 3.75–6.12). The TD rates were higher in female workers in most sectors of activity and occupations compared to men in the same sectors and with the same occupations. The results show the uneven impact of COVID-19 by occupation, with a higher

after request, from the social security website: https://sede.seg-social.gob.es/wps/portal/sede/sede/Ciudadanos/2022estadisticas/090316_c_ai?changeLanguage=en".

**Funding:** The author(s) received no specific funding for this work.

**Competing interests:** NO authors have competing interests.

rates in less qualified occupations (unskilled workers and laborers) *versus* the first categories of the table (directors, managers, technicians, and professionals).

## Conclusions

The results confirmed the high impact of COVID-19 on Spanish non-healthcare workers and it's inequalities. They also confirmed the potential use of TDs as an alternative source for epidemiological, public health surveillance and early warning of new emerging infections.

## Introduction

In response to the public health emergency and international pandemic, the Government of Spain enacted Royal Decree 463/2020, on March 14, declaring a state of alarm to manage the health crisis caused by COVID-19 [1]. To protect the citizens' health, contain the progression of the disease, and reinforce the public health system, this Royal Decree included temporary limitations on the free movement of people and preventive measures in the educational, work, commercial, recreational, and religious spheres. Certain measures were also taken to strengthen the National Health System throughout the country, ensure the supply of goods and services necessary for public health protection, food supply, energy supply and transport measures. Only those considered essential workers were allowed to work.

The most affected sectors in Europe (hotels, catering, transport and leisure) represent around 9% in the euro area, but their weight is higher in Spain (13%) or Italy (11%) [2]. Studies carried out in Spain mention that household consumption decreased by 12.4%, while public spending increased by 3.8%. The investment sector dropped by 12.4%; construction and capital goods also dropped by 14% and 13%, respectively [3].

With the aim of minimizing the social impact and facilitating the recovery of activity as soon as the health situation improved, the European Commission, in its communication of March 13, emphasized that the response should be coordinated, with support from institutions and the EU budget for measures taken in each member country. In Spain, Royal Decree-Laws 6/2020, of March 10, and 8/2020, of March 17, on urgent measures in the economic, social, and public health spheres [4, 5], expanded the previously implemented measures with an economic and social package oriented towards a triple objective. First, to reinforce the protection of workers, families, and vulnerable groups; second, to support the continuity of jobs and productive activity; and third, to strengthen the fight against the disease.

Among these measures, special mention should be made of temporary disability (TD) for (essential) workers who had to stay at home for health reasons [4]. Three situations were identified for workers protection: sick with COVID-19, in quarantine and observation due to close contact with cases, and vulnerable workers (cardiovascular disease, including hypertension, chronic lung disease, diabetes, chronic kidney failure, immunosuppression, cancer in the active treatment phase, severe chronic liver disease, morbid obesity (BMI>40), pregnancy and those over 60 years of age). The latter group (vulnerable workers) was not analysed to avoid bias in the study. At the beginning of the pandemic, it was difficult to know where COVID-19 infections were occurring. Initial information came from the outbreak notification system, which recorded the number of new outbreaks and the cases associated with them [6]. However, the identification of transmission chains was biased towards more closely monitored sectors, such as healthcare and sociosanitary centers, or towards those where traceability was easier, as in family or workplace outbreaks. Traceability in places where unknown individuals

converged was much more complicated, and these areas were underrepresented in outbreak information systems.

In previous studies, different papers have described the impact of COVID-19 in terms of TD in the healthcare and sociosanitary population [7–12]. However, it is important to highlight that the beginning of the second wave in Spain and later in Europe had its origin in Agriculture seasonal workers [6]. This shows the relevance of monitoring the work settings and the working and living conditions of workers to better understand the COVID-19 transmission mechanisms. Notwithstanding this, there are very few studies describing the magnitude of the impact on the non-healthcare working population [6, 13, 14]. Consequently, our current knowledge of the COVID-19 impact in these workers is still incomplete.

The objective of our research is to describe the impact of the pandemic on the Spanish Non-Healthcare Working Population (hereinafter referred to as NHWP) (excluding healthcare and sociosanitary personnel) during the first and second pandemic waves. This aims to evaluate the work settings where there may have been a higher circulation of the coronavirus. A comparative analysis of temporary incapacities due to COVID-19 in different economic activities and occupations, as well as the characteristics or sociodemographic aspects of the affected workers, is performed.

## Methodology

The cases under study were the processes of temporary disability due to COVID-19 (ICD-10 ES codes: B34.2, B97.21, and U07.1) and quarantine due to close contact with a COVID-19 case (ICD-10 ES code: Z20.828) from February 15 to September 17, 2020, provided by the National Institute of Social Security (INSS), by age, gender, economic activity and occupation.

We use the European Classification of Economic Activities (NACE) [15] for the analysis by economic activity and the International Classification of Occupations (ISCO) [16] for the analysis by occupation. These classifications provide good indicators of the working conditions of workers in a complementary way.

For the economic activity analysis, healthcare and sociosanitary activities were excluded (Q86, Q87, and Q88 codes from the NACE. For the occupation analysis, healthcare and personal care professionals were also excluded (codes 21, 56 and 57 of the ISCO).

The ratio between TD due to quarantine and TD due to illness, defined as the ratio between TD due to quarantine and TD due to illness in the last 7 days, was calculated. For every economic sector of activity the median and interquartile range (IQR), representing the difference between the third quartile (Q3) and the first quartile (Q1), were calculated.

$$IQR = Q3 - Q1$$

This allowed comparing the dispersion among sectors.

Population figures affiliated with social security, disaggregated at 1, 2, and 3 NACE digits, were used as denominators for rates calculations [17]. To calculate rates by occupation, figures of employed individuals at 1, 2, and 3 ISCO digits were used, provided by the Economically Active Population Survey [18].

The following rates of COVID-19 temporary disability processes were obtained as rates by [1] economic activity, and [2] occupation.

COVID-19 Temporary disability Rate by Economic Activity (TDREA) was defined as: Number of temporary disability processes (NTD) in a NACE code per 100,000 active affiliates to Social Security in that NACE code.

$$TDREA = \frac{NTD \text{ by NACE code} * 100,000}{\text{Number of active affiliates to Social Security by NACE code}}$$

COVID-19 Temporary disability Rate by Occupation (TDRO) was defined as: Number of temporary disability processes in an ISCO code per 100,000 employed individuals in that ISCO code.

$$TDRO = \frac{\text{NTD By ISCO code} * 100,000}{\text{Number of employed individuals by ISCO code}}$$

A differentiated analysis was conducted by gender, both for economic activity and occupation.

Helsinki standards were met. Ethics committee approval was not required because anonymised data are used, and the information corresponds to epidemiological surveillance data.

## Results

Between February 15, 2020, and September 17, 2020, 1,440,065 COVID-19-related TD processes were recorded in Social Security, of them 313,309 corresponded to healthcare and sociosanitary activities, and were excluded. Among the remaining cases, 74.4% (837,787) were due to quarantine from close contact with COVID-19 cases, and 25.6% (288,969) were due to illness. Of the TD processes, 54.6% (615,083) were observed in men (representing 54% of the studied working population), and 45.4% (511,672) in women (46% of that working population). The TD rate reached values of 5,465 quarantines and 1,878 illnesses per 100,000 women, and 4,883 quarantines and 1,690 illnesses per 100,000 men.

Regarding illness cases leading to TD, the average age was 42 years for both men and women. The highest number of COVID-19-related TD processes occurred between the ages of 30 and 59. However, the highest incidence rates of TD due to illness were observed in younger age groups, under 30, both in men and women (Table 1).

For the economic activity analysis, 46,970 out of 288,969 TD processes due to illness did not contain this information and were excluded from the analysis. The 99 TD processes corresponding to *Activities of extraterritorial organizations* (group U of the NACE) were also excluded due to the instability of their figures, resulting in 241,900 analyzed processes in our study. For the occupation analysis, 54,774 TD processes were eliminated for not containing information on occupation, 14,449 for belonging to occupations related to healthcare or sociosanitary services, and 39 *Armed Forces Occupations* processes. This resulted in a total of 219,707 cases included in this analysis. Table 1 shows the percentage of workers, the number, percentage, and rate of COVID-19-related TD by age group, economic activity, and occupation at 1-digit NACE and ISCO08, for women and men.

Fig 1 illustrates the distribution of TD rates per 100,000 affiliated individuals for COVID-19 illness (in red) and quarantine (in orange), according to economic activity at 2-digit NACE (excluding healthcare and sociosanitary activities), for the period from February 15, 2020, to September 17, 2020. The ratio between temporary disability due to quarantine and temporary disability due to illness (close contacts and cases) had a median of 2.6 quarantines per case (IQR: 2.2–3.6) throughout the studied period, with the distribution of values shown in Fig 2 for the specified economic activities.

In *Postal and Courier activities*, the median was 4.68 quarantines per case (IQR: 3.75–6.12) during the studied period. The *Manufacture of electrical equipment* and the *Waste collection, treatment, disposal activities and materials recovery* showed values close to 4: 3.88 (IQR: 2.89–5.28) and 3.79 (IQR: 2.85–4.39), respectively. The median was 3.5 quarantines per case in the *Repair of computers and household goods* (IQR: 2.8–5), similar to *Retail Trade except motor vehicles and motorcycles* (3.43, IQR: 2.66–4.13) and *Repair and installation of machinery and equipment* (3.42, IQR: 2.23–4.06) (Fig 2). *Food and Beverage Services Activities*, *Wholesale*

**Table 1. Distribution of the non-health Spanish workers, number, percentage and rate of COVID-19 temporary disability (TD) per 100,000, by age, economic activity and occupation, by sex.** Spain, February 15th to September 17th, 2020.

| | Female | | | | Male | | | |
|---|---|---|---|---|---|---|---|---|
| | % FW | NTD | % TD | TDR | % MW | NTD | % TD | TDR |
| **Age groups** | | | | | | | | |
| 16–19 | 0.63 | 1,135 | 0.87 | **2,006** | 0.77 | 1,522 | 0.96 | **1,921** |
| 20–29 | 13.50 | 20,376 | 15.57 | **1,681** | 13.35 | 24,965 | 15.79 | **1,820** |
| 30–39 | 23.77 | 32,279 | 24.67 | **1,513** | 23.00 | 38,830 | 24.56 | **1,643** |
| 40–49 | 30.54 | 40,066 | 30.62 | **1,461** | 30.69 | 47,134 | 29.81 | **1,495** |
| 50–59 | 23.68 | 29,248 | 22.35 | **1,376** | 24.17 | 35,580 | 22.50 | **1,433** |
| 60–65 | 6.71 | 7,297 | 5.58 | **1,211** | 6.78 | 9,267 | 5.86 | **1,331** |
| > 65 | 1.17 | 443 | 0.34 | **421** | 1.24 | 826 | 0.52 | **651** |
| **Overall Rate** | 100 | 130,844 | 100 | **1,878** | 100 | 158,124 | 100 | **1,690** |
| **Economic Activity** | | | | | | | | |
| A: Agriculture, Forestry, and Fishing | 1.4 | 2,380 | 2.13 | **2,529** | 2.59 | 4,477 | 3.44 | **1,846** |
| B: Mining and quarrying | 0.0 | 25 | 0.02 | **957** | 0.19 | 208 | 0.16 | **1,153** |
| C: Manufacturing | 8.0 | 10,258 | 9.17 | **1,842** | 16.24 | 27,501 | 21.15 | **1,810** |
| D: Electricity, gas, steam and air conditioning supply | 0.1 | 82 | 0.07 | **963** | 0.29 | 325 | 0.25 | **1,187** |
| E: Water supply; sewerage; waste management and remediation activities | 0.4 | 519 | 0.46 | **1,806** | 1.25 | 2,054 | 1.58 | **1,762** |
| F: Construction | 2.0 | 1,300 | 1.16 | **950** | 12.03 | 13,908 | 10.69 | **1,236** |
| G: Wholesale and Retail Trade; Repair of Motor Vehicles and Motorcycles | 22.3 | 23,369 | 20.89 | **1,504** | 17.68 | 19,999 | 15.38 | **1,209** |
| H: Transporting and Storage | 2.7 | 4,219 | 3.77 | **2,224** | 7.98 | 11,591 | 8.91 | **1,553** |
| I: Accommodation and food service activities | 11.8 | 9,327 | 8.34 | **1,138** | 7.94 | 7,581 | 5.83 | **1,020** |
| J: Information and communication | 2.8 | 2,041 | 1.82 | **1,044** | 4.07 | 3,611 | 2.78 | **949** |
| K: Financial and Insurance Activities | 2.9 | 2,885 | 2.58 | **1,445** | 1.94 | 2,090 | 1.61 | **1,152** |
| L: Real Estate Activities | 1.2 | 678 | 0.61 | **842** | 0.71 | 544 | 0.42 | **817** |
| M: Professional, Scientific, and Technical Activities | 7.6 | 5,627 | 5.03 | **1,057** | 5.68 | 4,462 | 3.43 | **840** |
| N: Administrative and Support Service Activities | 11.0 | 19,008 | 16.99 | **2,484** | 7.18 | 14,263 | 10.97 | **2,124** |
| O: Public Administration and Defence; Compulsory Social Security | 8.3 | 10,267 | 9.18 | **1,774** | 5.72 | 10,092 | 7.76 | **1,887** |
| P: Education | 10.4 | 5,184 | 4.63 | **717** | 3.99 | 2,849 | 2.19 | **764** |
| R: Arts, entertainment and recreation | 2.1 | 1,402 | 1.25 | **957** | 2.12 | 1,780 | 1.37 | **899** |
| S: Other services activities | 4.9 | 3,274 | 2.93 | **957** | 2.10 | 1,853 | 1.42 | **943** |
| T: Activities of households as employers; undifferentiated goods—and services—producing activities of households for own use | 0.2 | 10,004 | 8.94 | **76,663** | 0.31 | 863 | 0.66 | **2,990** |
| **Overall Rate** | 100 | 111,849 | 100 | **1,605** | 100 | 130,051 | 100 | **1,390** |
| **Occupation** | | | | | | | | |
| A: Directors and Managers | 3.4 | 2,245 | 2.3 | **845** | 4.7 | 5,131 | 4.1 | **1,061** |
| B: Health and Education Science and Intellectual Professionals | 9.8 | 2,674 | 2.8 | **351** | 3.5 | 1,375 | 1.1 | **383** |
| C: Other Science and Intellectual Professionals | 10.6 | 5,467 | 5.7 | **662** | 10.1 | 7,966 | 6.5 | **781** |
| D: Technicians; Support Professionals | 10.7 | 8,059 | 8.4 | **968** | 13.1 | 14,006 | 11.3 | **1,044** |
| E: Office Workers Not Dealing with the Public | 8.9 | 6,797 | 7.1 | **977** | 4.0 | 4,951 | 4.0 | **1,200** |
| F: Office Workers Dealing with the Public | 8.7 | 10,325 | 10.8 | **1,522** | 2.5 | 4,340 | 3.5 | **1,680** |
| G: Workers in Catering and Trade Services | 19.3 | 20,040 | 20.9 | **1,334** | 10.7 | 13,093 | 10.6 | **1,199** |
| H: Personal Services Workers (only CNO 58) | 3.6 | 3,847 | 4.0 | **1,359** | 1.8 | 3,306 | 2.7 | **1,844** |
| I: Security and Safety Service Workers | 0.7 | 1,439 | 1.5 | **2,662** | 3.6 | 6,695 | 5.4 | **1,797** |
| J: Skilled Workers in Agriculture, Livestock, Forestry, and Fishing | 1.0 | 653 | 0.7 | **850** | 3.4 | 1,944 | 1.6 | **564** |
| K: Skilled Construction Workers, Except Machine Operators | 0.2 | 400 | 0.4 | **2,735** | 8.4 | 9,866 | 8.0 | **1,153** |
| L: Skilled Workers in Manufacturing Industries, Except Facility and Machine Operators | 1.9 | 4,616 | 4.8 | **3,089** | 11.2 | 16,172 | 13.0 | **1,407** |
| M: Operators of Fixed Installations and Machinery, and Assemblers | 2.1 | 1,783 | 1.9 | **1,070** | 4.0 | 4,526 | 3.7 | **1,118** |
| N: Drivers and Operators of Mobile Machinery | 0.5 | 514 | 0.5 | **1,404** | 9.0 | 10,043 | 8.1 | **1,097** |

(*Continued*)

**Table 1.** (Continued)

| | Female | | | | Male | | | |
|---|---|---|---|---|---|---|---|---|
| | % FW | NTD | % TD | TDR | % MW | NTD | % TD | TDR |
| O: Unskilled Workers in Services (except Transportation) | 15.0 | 20,919 | 21.8 | **1,798** | 2.7 | 7,673 | 6.2 | **2,773** |
| P: Labourers in Agriculture, Fishing, Construction, Manufacturing Industries, and Transportation | 3.5 | 6,002 | 6.3 | **2,173** | 7.3 | 12,729 | 10.3 | **1,706** |
| **Overall Rate** | 100 | 95,780 | 100 | **1,231** | 100 | 123,927 | 100 | **1,212** |

FW: Female Workers//MW: Male Workers// NTD: Number of Temporary Disabilities // TDR Temporary Disability Rate

*Trade*, *Education*, and *Building Construction* presented medians slightly above the average (2.7, 2.69, 2.68, and 2.53, respectively), while in *Telecommunications* and *Activities of households as employers of domestic personnel*, they were below the average, at 2.09 (IQR: 1.69–3) and 1.82 (IQR: 1.53–2.42). Both the *Repair of computers*, *personal and household goods*, and the *Manufacture of computer*, *electronic*, *and optical products* showed greater dispersion of values (Fig 2).

*Activities of Households as Employers*, *Agriculture, livestock, forestry, and fishing*, *Administrative and support service* activities, *Transportation and storage*, *Manufacturing industry*, *Water supply*, *sanitation activities*, *waste management*, *and decontamination activities*, and *Public administration* showed TD rates above the overall rate, and higher rates in women compared to men, except in the case of the *Public administration*. The first of these activities recorded a rate 48 times higher than the average in women and 26 times higher in women than in men. Along with *Public Administration*, only *Construction*, *Extractive Industries*, *Supply of electricity*, *gas*, *steam*, *and air conditioning*, and the *Education* sector, showed higher TD rates in men (Table 1).

The analysis of economic activity at 2 and 3 NACE digits (Table 2 and S1 Table) allows identifying higher rates in women in the *Temporary employment agency activities*, *Waste collection*, and *Processing and preservation of meat and meat products*. The rate for *Temporary employment agency activities* was 19 times higher than the average in women. *Activities in call centers* and *the Manufacture of pharmaceutical products* recorded similar rates in men and women, with the former being the activity with the highest rate in men (Table 2). *The Processing and preservation of meat and meat products* had the second-highest rate in men.

In the analysis of TD by occupation, *Skilled female workers in manufacturing industries and construction*, *Female workers in Security and Safety Service*, and *Female laborers in agriculture*, *fishing*, *construction*, *manufacturing*, *and transport*, had the highest TD rates. Moreover, these rates were higher than the same rates observed in men. For men, the highest TD rate was for *Unskilled workers in services (except transport)*, followed by *workers in Security and safety service* and *labourers in agriculture*, *fishing*, *construction*, *manufacturing*, *and transport* (Table 1).

When descending into the analysis at 2 and 3 digits (Table 3 and S2 Table), *Other craft and related workers* and *Other elementary workers* stood out as the occupational subgroups with the highest TD rates, both for men and women. Females in *Other craft and related workers* had rate values 55 times above their average and 1.5 times higher than that obtained for men. In the case of men, this rate was 37 times higher than their overall average by occupation. Females in *Other elementary workers* had a rate 12 times higher than their overall average and 1.7 times higher than that of men. In men, this rate was 7 times higher than their overall average by occupation.

In third place, for women, were *Transport and Storage Labourers (especially shelf fillers)*, followed by *Other personal service workers (especially companions and valets)*. In men, *Messengers*,

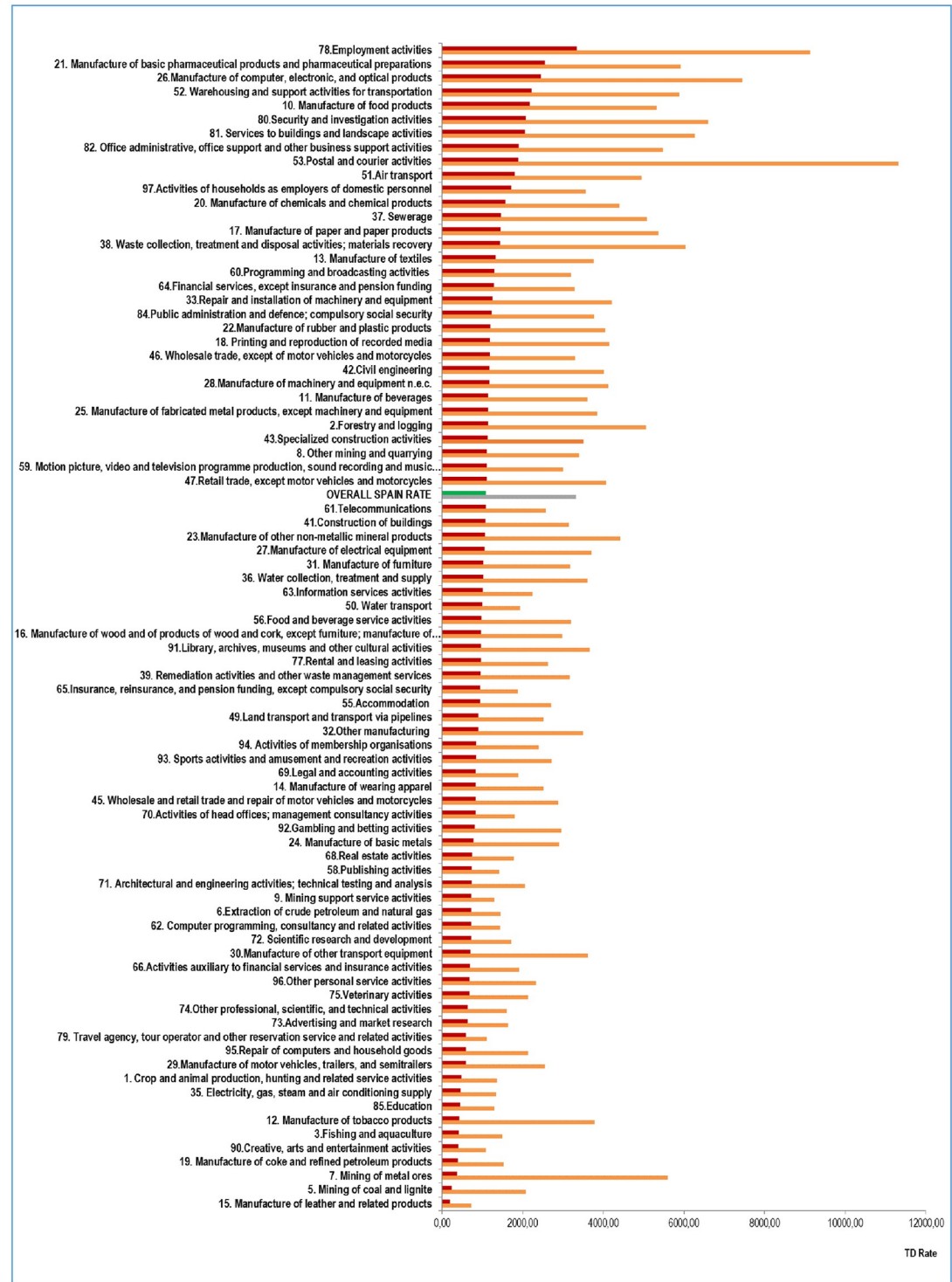

**Fig 1. Temporary disability (TD) per 100,000 affiliated individuals in the non-health Spanish workers, due to COVI D-19 illness (in red) and quarantine (in orange) by economic activity.** Spain, February 15th to September 17th, 2020.

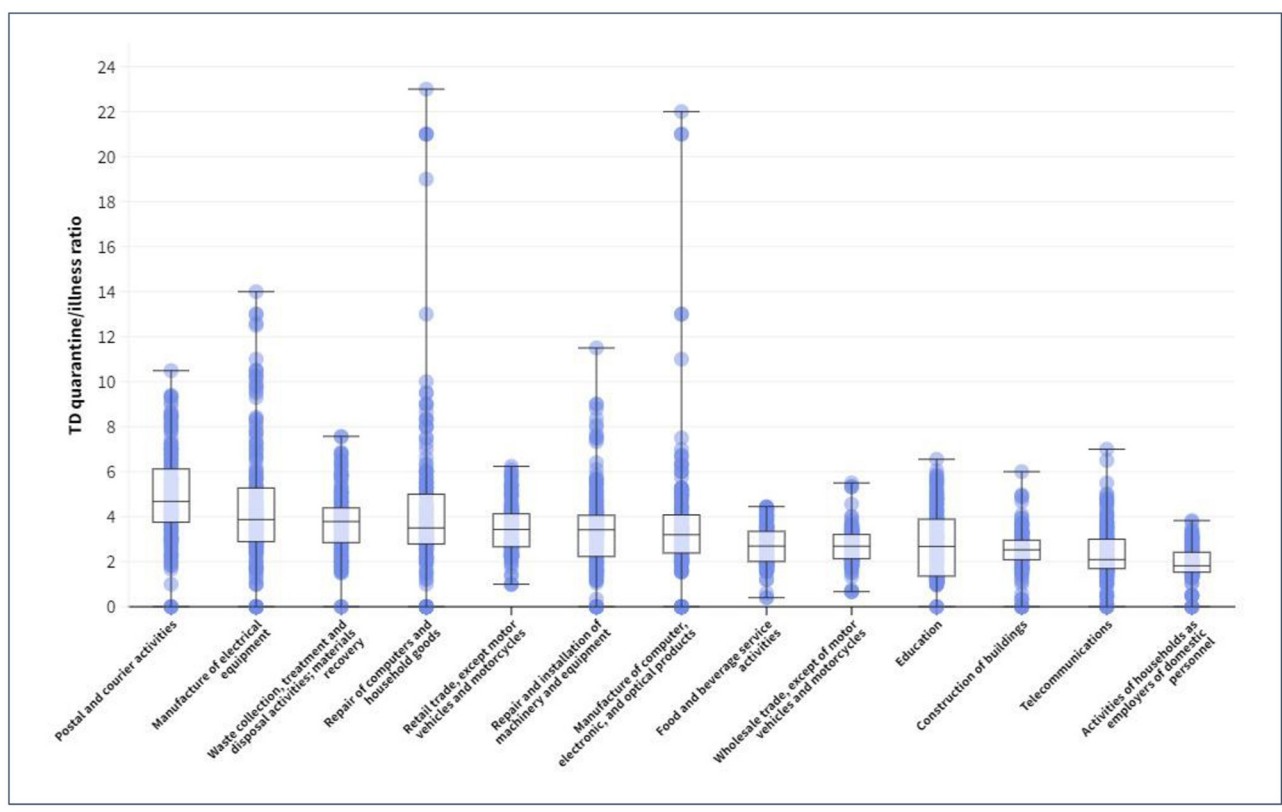

**Fig 2. Distribution of the ratio of temporary disability due to quarantine and to COVI D-19 illness (median, quartiles of the data and outliers), by the indicated economic activities.** Spain, February 15th to September 17th, 2020.

*Package Deliverers and Luggage Porters*, and *the Cleaners and Helpers* had the next two highest TD rates. The fifth place is occupied by *Food Processing (especially butchers, fishmongers and related food preparers)*, both in women and men, with the rate observed in women higher than that obtained for men (Table 3).

## Discussion

The results revealed the high impact in terms of sick leave associated with COVID-19 in Spanish workers and important differences by gender, economic activity and occupation that have characterized this impact. The TD rate was 26 and 9.5 times higher for women than for men in the *Activities of households as employers of domestic personnel* and in the *Temporary employment agency activities*, respectively. *Waste Collection*, *Activities of call centers*, and *Processing and preserving of meat products*, were also sectors with high risk of exposure for both genders. Taking into account the workers occupation, the *less qualified* presented the highest rates.

TD already was proven as a useful tool to complement the traditional surveillance system [7, 19]. It is closely correlated with the evolution of the Accumulated Incidence collected by the National Epidemiological Surveillance Network in individuals aged 18 to 65 years, even serving as an advanced indicator of the incidence at seven days, as shown previously in other sectors [7, 8].

The average age of all TD cases was 42 years, both in men and women, similar to findings in other studies conducted on the working population [20]. The highest number of COVID-19

**Table 2. Temporary disability due to COVID-19, per 100,000 workers, in economic activities with the highest prevalence (excluding the healthcare and social healthcare sectors), by gender.** Spain, February 15th to September 17th, 2020.

| Women | | | | Men | | | |
|---|---|---|---|---|---|---|---|
| Economic activity | NTD | FW | TDREA | Economic activity | NTD | MW | TDREA |
| 97. Activities of households as employers of domestic personnel | 10,004 | 13,049 | **76,663** | 822. Activities of call centres | 865 | 16,925 | **5,111** |
| 782. Temporary employment agency activities | 2,586 | 8,295 | **31,175** | 101. Processing and preserving of meat | 3,992 | 88,278 | **4,522** |
| 381. Waste collection | 305 | 4,602 | **6,629** | 601. Radio broadcasting | 222 | 6,176 | **3,595** |
| 822. Activities of call centres | 2,038 | 39,718 | **5,131** | 26. Manufacture of computer, electronic and optical products | 669 | 20,497 | **3,264** |
| 101. Processing and preserving of meat and production of meat products | 1,777 | 35,625 | **4,988** | 782. Temporary employment agency activities | 3,855 | 116,417 | **3,311** |
| 26. Manufacture of computer, electronic and optical products | 390 | 8,746 | **4,459** | 21. Manufacture of basic pharmaceutical products and pharmaceutical preparations | 826 | 26,769 | **3,086** |
| 52. Warehousing and support activities for transportation | 2,048 | 62,232 | **3,291** | 97. Activities of households as employers of domestic personnel | 863 | 28,865 | **2,990** |
| 21. Manufacture of basic pharmaceutical products and pharmaceutical preparations | 815 | 26,395 | **3,088** | 381. Waste collection | 1,207 | 43,962 | **2,746** |
| 563. Beverage serving activities | 2,754 | 90,510 | **3,043** | 52. Warehousing and support activities for transportation | 3,344 | 138,673 | **2,411** |
| 81. Services to buildings and landscape activities | 10,757 | 401,153 | **2,682** | 172. Manufacture of paper and paper products (including sanitary goods and of toilet requisites) | 484 | 21,193 | **2,284** |
| 01.Agriculture | 2,338 | 89,980 | **2,598** | 103. Processing and preserving of fruit and vegetables | 559 | 26,196 | **2,134** |
| 103. Processing and preserving of fruit and vegetables | 536 | 21,356 | **2,510** | 53. Postal and courier activities | 1,037 | 50,429 | **2,056** |
| 471. Retail sale in non-specialised stores (with food, beverages or tobacco predominating) | 8,835 | 366,029 | **2,414** | 20. Manufacture of chemicals and chemical products | 1,396 | 69,148 | **2,019** |
| 601. Radio broadcasting | 156 | 6,690 | **2,332** | 81. Services to buildings and landscape activities | 4,085 | 204,537 | **1,997** |
| 53. Postal and courier activities | 773 | 35,349 | **2,187** | 801. Private security activities | 2,214 | 111,713 | **1,982** |
| 801. Private security activities | 572 | 28,086 | **2,037** | 84. Public administration and defence; compulsory social security | 10,092 | 534,867 | **1,887** |
| 75. Veterinary activities | 342 | 17,990 | **1,901** | 01.Agriculture | 4,005 | 218,898 | **1,830** |
| 84. Public administration and defence; compulsory social security | 10,267 | 578,787 | **1,774** | 22. Manufacture of rubber and plastics products | 1,226 | 69,008 | **1,777** |

NTD: Number of temporary disabilities //FW: Female workers// MW: Male Workers// TDREA: Temporary disability rate by economic activity

TD cases occurs in middle-aged adults, however, the highest incidence rates of TD due to illness were recorded in younger age groups. The higher impact on the younger population might be influenced by increased exposure for these workers. From very early stages in the pandemic, it was observed that the severity of the disease was greater in older individuals [21–23], and they were likely protected from exposure to some extent.

Regarding gender, data from our study indicates a greater impact of COVID-19 on Spanish female workers in most sectors of activity and occupations compared to men in the same sectors and with the same occupations. For example, *Temporary employment agency activities*, recorded a rate 9.5 times higher in women than in men, or 2.5 times higher in *Waste Collection*. *Other craft and related female workers* had a rate 1.5 times higher than men in the same positions, and *Other elementary occupations* had a rate 1.7 times higher for female than their male counterparts. Regarding *Temporary employment agency activities*, several high risk occupations were identified for women, such as *Agricultural labourers, Call centers operators, Fruit and vegetable preservers and sellers*, whereas *Transport and manufacturing labourers*, Craft workers and *Butchers*, were high risk occupations for men. One possible explanation is

**Table 3. Temporary disability due to COVID-19, per 100,000 workers, in occupations with the highest prevalence (excluding healthcare and social healthcare personnel), by gender.** Spain, February 15th to September 17th, 2020.

| Females | | | | Males | | | |
|---|---|---|---|---|---|---|---|
| Occupation | NTD | FW | TDRO | Occupation | NTD | MW | TDRO |
| 754. Other Craft and Related Workers | 1,102 | 1,639 | **67,228** | 754. Other Craft and Related Workers | 3,147 | 7,069 | **44,518** |
| 9621. Other Elementary Workers | 936 | 6,456 | **14,499** | 9621. Other Elementary Workers | 1,570 | 18,455 | **8,507** |
| 933 Transport and Storage Labourers (especially shelf fillers) | 1,112 | 26,636 | **4,175** | 9621. Messengers, Package Deliverers and Luggage Porters | 1,944 | 30,729 | **6,326** |
| 516. Other Personal Services Workers (especially companions and valets) | 666 | 18,213 | **3,657** | 91 Cleaners and Helpers | 6,475 | 103,905 | **6,232** |
| 751. Food Processing (especially butchers, fishmongers and related food preparers) | 2,700 | 80,920 | **3,337** | 751. Food Processing (especially butchers, fishmongers and related food preparers) | 3,660 | 136,063 | **2,690** |
| 9621. Messengers, Package Deliverers and Luggage Porters | 487 | 15,357 | **3,171** | 932. Manufacturing Labourers | 2,523 | 95,559 | **2,640** |
| 335. Government regulatory associate professionals and 541. Protective service workers (primarily police officers) | 633 | 20,156 | **3,140** | 335. Regulatory government associate professionals and protective service workers (primarily police officers) | 3,093 | 123,873 | **2,496** |
| 92. Agricultural labourers | 2,397 | 77,021 | **3,112** | 311. Physical and engineering science technicians (especially electrical and mechanical engineering technicians) | 2,731 | 106,492 | **2,565** |
| 541. Security guards and Protective Services Workers Not Elsewhere Classified | 870 | 30,938 | **2,812** | 515. Building and housekeeping supervisors | 2,273 | 96,523 | **2,355** |
| 422. Client Information Workers, primarily telephone switchboard operators | 2,921 | 134,060 | **2,179** | 422. Client Information Workers, primarily telephone switchboard operators | 1,287 | 57,840 | **2,225** |
| 91 Cleaners and Helpers | 12,865 | 594,489 | **2,164** | 593. Firefighters | 648 | 31,625 | **2,049** |
| 513. Waiters | 4,805 | 270,305 | **1,778** | 541. Security guards and Protective Services Workers Not Elsewhere Classified (5414 & 5419) | 3,297 | 168,664 | **1,954** |
| 523. Cashiers and ticket clerks | 2,293 | 132,997 | **1,724** | 960. Construction labourers | 2,353 | 123,896 | **1,899** |
| | | | | 9311.Building Construction Labourers | | | |
| 753. Garment and related trades workers | 359 | 24,778 | **1,449** | 4221. Client Information Workers Not Elsewhere Classified | 2,505 | 140,498 | **1,783** |
| 816 & 819. Food and Related Products and Other Stationary Plant and Machine Operators | 839 | 59,015 | **1,422** | 783. Workers in textile, clothing, leather, and footwear industries | 417 | 26,318 | **1,584** |
| 970. Labourers in manufacturing industries | 1,742 | 122,905 | **1,417** | 511 and 931. Cooks and kitchen helpers | 2,747 | 174,097 | **1,578** |
| 450. Administrative employees with public-facing tasks not classified under other headings | 6,290 | 452,141 | **1,391** | 98. Transportation labourers, unloaders, and stock replenishers | 3,960 | 251,459 | **1,574** |
| 511 y 931. Cooks and kitchen helpers | 3,169 | 238,544 | **1,329** | 550. Cashiers and ticket sellers (except banks) | 500 | 33,157 | **1,508** |
| 981. Transportation labourers and unloaders | 505 | 40,359 | **1,251** | 951 y 953. Agricultural labourers | 3,385 | 240,699 | **1,406** |
| 910. Domestic employees | 4,988 | 416,177 | **1,199** | 513. Waiters and Bartenders | 3,681 | 277,244 | **1,326** |
| 441. Information and receptionists (except from hotels) | 677 | 58,210 | **1,163** | 9611–9613 Garbage and recycling collectors, refuse sorters, sweepers and related labourers | 688 | 60,126 | **1,144** |

NTD: Number of temporary disabilities //FW: Female workers// MW: Male Workers// TDRO: Temporary disability rate by occupation

workplace segregation, where women are confined to a narrower range of occupations (horizontal segregation) and lower-ranking positions (vertical segregation), leading to differential exposure to risk factors, working conditions, and living conditions [24, 25]. It is increasingly accepted in the literature that estimating occupational risks by gender without considering the unequal distribution and participation of men and women in the workforce leads to a misassessment of professional risks. These observed gender inequalities should be the subject of more specific studies with appropriate designs to confirm this hypothesis. Gender aspects are highly relevant, and reliable data should form the basis for prevention policies, intervention strategies, and the subsequent evaluation and monitoring of their effectiveness [26]. According

to our results, ignoring the differential ways in which female workers become ill may lead to discrimination that inhibits preventive policies for diseases that affect women more than men.

The results also show the uneven impact of COVID-19 by the economic activity of the company or the occupation of the worker. The greatest impacts were observed in sectors such as *Activities of households as employers of domestic personnel*, *Temporary employment agency activities*, *Waste Collection*, *Activities of call centres*, and *Processing and preserving of meat and meat products*, essential sectors that have not interrupted their on-site work at any point in the evolution of the epidemic, either because telework is not possible (or more challenging), or because the activity has been more intense due to the impact of the pandemic [27].

In this group of activities, it is worth noting the TD rate for domestic employees (NACE T and 97), which is unusually high, especially in women, likely due to an overestimation resulting from not including all workers in the sector in the denominator. It should be noted that affiliation data does not include special regimes, which may be numerous for this group [28]. On the other hand, we have excluded and do not show the rates corresponding to *Activities of extraterritorial organizations* (NACE U) and *Armed Forces Occupations* (ISCO Q), as they present extremely small TD values, 99 and 39 processes, respectively, likely due to a notification bias [28].

Sectors such as *Education* or *Food and beverage service activities* show a lower impact. In the case of *Education*, during the second wave, TD increased from late August and early September, possibly indicating the massive screenings that many authorities carried out on teachers before the start of the school year. In the case of *Food and beverage service activities*, it should be considered that a large number of establishments were closed during both the first and second waves, reducing the professional exposure of this group to COVID-19 and the incidence of TD [29, 30].

In the surveillance of COVID-19 outbreaks in Spain, established in June 2020 as part of the *Early Detection, Surveillance, and Control Strategy for COVID-19*, the second group of outbreaks was related to work environments. Until July 31, 2020, 108 outbreaks of occupational origin were recorded, representing 20% of all active outbreaks. Among them, outbreaks related to temporary workers in the horticultural sector and workers in slaughterhouses or meat processing plants were the most frequent [6]. In our study, *Agriculture*, as a major NACE group, recorded a high rate of TD due to infection, both for men and women, in line with the reported outbreaks in the sector and also with similar findings internationally [31–33]. The sector of *Processing and preserving of meat and meat products*, on the other hand, presented the second-highest rate in men and the fifth in women. We can define these sectors as more vulnerable, understanding vulnerability as a higher risk of infection due to greater exposure, inability to telework, delay in diagnosis and identification of contacts, or greater difficulty in following prevention measures, as well as isolation or quarantine, due to their living and working conditions. All of these factors would explain the high rates found in our study.

Regarding occupation, the results show the uneven impact of COVID-19, with a gradient from lower to higher affectation from the first categories of the table (*directors, managers, technicians, and professionals*) to the last positions: *unskilled workers and laborers*. Differences exist not only by age or gender but also by income or the type of work. Less qualified occupations present higher rates. They do not telework. They are essential occupations that do not allow for remote work. If there is an infection, they cannot isolate themselves at home, with only one bathroom or even several families living together. Also, with more education and socioeconomic status, they have a higher ability to receive, integrate, and comply with messages. There is evidence that socioeconomic status, sociodemographic characteristics, as well as the social determinants of health to which these groups are exposed, play an important role, generating

socioeconomic inequalities associated with a higher risk of infection, as observed in outbreaks affecting slaughterhouses and seasonal workers [6, 34–37].

In our study, we have confirmed these findings obtained in other studies and verified the different distribution in men and women described earlier. As observed in the analysis by economic activity of the industry, the occupation analysis showed higher rates in women than in men in many of them. *Other craft and related workers* stood out as the occupational group with the highest TD rates, and within this group, the subgroup of *Workers in the food, beverage, and tobacco industry (especially butchers and workers in the meat industry)*, was identified as having the highest proportion of TD, for both men and women. These results coincide with those provided by the analysis by economic activity and confirm the high impact of the pandemic on the meat industry. At European level, the food production sector, which includes food processing and production, is the second-highest in terms of cases after the health sector [33]. In the USA, there is also evidence of a higher risk of infection in workers involved in the food production chain [38, 39].

Other elementary occupations such as *Messengers*, *Package Deliverers and Luggage Porters*, *Cooks and kitchen helpers*, *Refuse workers*, and *Cleaning staff*, recorded a higher impact in men. In this regard, building cleaning personnel were identified as high-risk personnel for COVID-19 infection in the USA [40] and Germany [41]. Regarding cooks and kitchen assistants, some studies also identify occupations in the restaurant and hotel sector among the most affected, for example, in Catalonia [14], the USA [40], and Norway [42], which we have not confirmed in our study.

Finally, it is worth noting other occupations such as *Regulatory government associate professionals (primarily police officers)*, the *Security guards and Protective Services Workers*, and *Firefighters*. The in-person nature of these occupations with the impossibility of remote work would have increased the possibility of infection for these groups.

Our study has certain limitations. The first of them is the nature of the analyzed data, which are collected for purposes other than health event surveillance. Its main objective is of economic nature, as social security provision for the economic protection of workers, so the data may lack information of interest in the analysis for health purposes. The TD registry does not include people who do not work or work in the informal economy and also get sick but do not take TD. Or, conversely, the public health protection character attributed to these COVID-19 TD may have increased the number of sick individuals to a greater extent than the number of workers included in the denominator. Likewise, the lack of a unique identifier has not allowed distinguishing successive TD granted to the same worker, which may lead to an overestimation of incidence.

Another limitation has to do with the denominators of the rates used in this study, the figures of the population affiliated with social security, for the calculation of rates by economic activity, and the figures of employed provided by the Economically Active Population Survey (EPA), for the rates by occupation. Although the data provided by the EPA does not exactly correspond to the data of affiliation to social security, there are studies that have shown the good correlation between the rates obtained with both sources [43]. Nevertheless, the values of some rates may be distorted. This, together with the non-inclusion of special regimes in affiliation data, is an explanation for the atypical values of the rates obtained for domestic employees.

Another limitation is the impossibility of identifying the place where the infection occurred. Associating infections with places and jobs, that is, with certain working conditions, does not incorporate living conditions, commutes to workplaces, and other socioeconomic and environmental determinants, which are relevant to characterize the pandemic and were relevant in outbreaks of slaughterhouses and seasonal workers, for example.

As a strength of the study, it is one of the most extensive conducted to date, describing the impact of COVID-19 in terms of TD or, in other words, the magnitude of the impact on the non-healthcare working population in Spain.

## Conclusions

In conclusion, the TD system is a valuable supplementary source of information for analyzing the impact of COVID-19 in different work settings. In the first and second waves of COVID-19 in Spain, the impact was greater in women and more in young people in most activities and occupations. A gradient was observed in the incidence in different occupations, being higher in less qualified workers. These results allow identifying the economic activity sectors and occupations most affected by COVID-19, which can help guide future prevention policies in work environments.

It would be important to do some targeted research on the most affected groups in order to determine specifically what other factors might have influenced the increased incidence in some groups of non-health workers.

## Supporting information

**S1 Table. Temporary disability due to COVID-19, per 100,000 workers, by economic activity (excluding the healthcare and social healthcare sectors), by sex.** Spain, February 15th to September 17th, 2020.
(DOCX)

**S2 Table. Temporary disability due to COVID-19, per 100,000 workers, by occupation (excluding the health and social care personnel), by sex.** Spain, February 15th to September 17th, 2020.
(DOCX)

## Acknowledgments

We would like to thank the National Institute of Social Security for providing us with data on TD.

## Author Contributions

**Conceptualization:** Montserrat García Gómez.

**Data curation:** Dante R. Culqui L., Alín Manuel Gherasím, Sofía Escalona López, Montserrat García Gómez.

**Formal analysis:** Dante R. Culqui L., Sofía Escalona López, Jesús Oliva Domínguez, Montserrat García Gómez.

**Investigation:** Dante R. Culqui L., Alín Manuel Gherasím, Sofía Escalona López, Montserrat García Gómez.

**Methodology:** Dante R. Culqui L., Alín Manuel Gherasím, Jesús Oliva Domínguez, Montserrat García Gómez.

**Project administration:** Montserrat García Gómez.

**Supervision:** Montserrat García Gómez.

**Validation:** Dante R. Culqui L., Montserrat García Gómez.

**Visualization:** Dante R. Culqui L., Alín Manuel Gherasím, Sofía Escalona López, Jesús Oliva Domínguez.

**Writing – original draft:** Dante R. Culqui L., Montserrat García Gómez.

**Writing – review & editing:** Alín Manuel Gherasím, Sofía Escalona López, Jesús Oliva Domínguez, Montserrat García Gómez.

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
