## [Decision Letter · Decision Letter 0]

31 May 2024

PONE-D-24-01071Health impact of the COVID-19 in Spanish workers by gender: use of sickness absence for surveillance.PLOS ONE

Dear Dr. Culqui L.,

Thank you for submitting your manuscript to PLOS ONE. After careful consideration, we feel that it has merit but does not fully meet PLOS ONE’s publication criteria as it currently stands. Therefore, we invite you to submit a revised version of the manuscript that addresses the points raised during the review process.

We look forward to receiving your revised manuscript.

Kind regards,

Luca Valera

Academic Editor

PLOS ONE

2. Please amend your manuscript to include your abstract after the title page.

3. In the online submission form you indicate that your data is not available for proprietary reasons and have provided a contact point for accessing this data. Please note that your current contact point is a co-author on this manuscript. According to our Data Policy, the contact point must not be an author on the manuscript and must be an institutional contact, ideally not an individual. Please revise your data statement to a non-author institutional point of contact, such as a data access or ethics committee, and send this to us via return email. Please also include contact information for the third party organization, and please include the full citation of where the data can be found.

Reviewers' comments:

Reviewer's Responses to Questions

**Comments to the Author**

1. Is the manuscript technically sound, and do the data support the conclusions?

Reviewer #1: Yes

Reviewer #2: Partly

Reviewer #3: Partly

2. Has the statistical analysis been performed appropriately and rigorously? 

Reviewer #1: Yes

Reviewer #2: No

Reviewer #3: No

3. Have the authors made all data underlying the findings in their manuscript fully available?

Reviewer #1: Yes

Reviewer #2: Yes

Reviewer #3: No

4. Is the manuscript presented in an intelligible fashion and written in standard English?

Reviewer #1: Yes

Reviewer #2: No

Reviewer #3: Yes

5. Review Comments to the Author

Reviewer #1: Peer-Review for PLOS ONE

Date of the review

• 31.01.2024

Title of the manuscript

• "Health impact of the COVID-19 in Spanish workers by gender: use of sickness absence for surveillance.”

Manuscript Identifier:

• PONE-D-24-01071

• First draft

Authors of the draft:

• Not blinded to the reviewer

Type of the manuscript

• primary study

Reviewer's report

Thank you for the opportunity to review the manuscript.

The aim of the study is describing the impact of the pandemic on the Spanish non-healthcare workers measured through health-related workplace absences based on disability data provided by the National Social Security. Temporary disability rates due to Covid-19 by economic activity and be occupation and ratios of the distribution of temporary disability due to quarantine and to Covid-19-illness are used to describe risks.

Scientific background, the aim of the study, hypothesis, statistical methods, results and the discussion are understandable, well and compact written and provide an impressive and good scientific view to the topic.

The reviewers have no additional considerations, recommendations or comments on the title, abstract, keywords, introduction, objective, methods, tables and the figure, the discussion and other aspects (references, acknowledgements etc.). There are no points left. The reviewer recommends the publication of the manuscript in PLOS ONE.

Reviewer #2: In this study the authors have investigated the share of temporary disability among non-healthcare working population in the two first waves of the pandemic in Spain. Temporary disability is defined as persons either in quarantine or sick due to covid-19 infection. The authors have investigated this based on gender, age, and occupation.

After carefully reading the paper, the general feeling is that the paper seems a bit unfished. There are several spelling errors, repetitive sentences, and the paper is missing a “story”. I therefore do not believe the paper is ready to be published yet.

Introduction:

1) The introduction is working fine. However, I miss a specific knowledge gap this paper is trying to fill. Why is this important and needed? Why is there a need to look at both economic activity and occupation? What new and needed information does this bring?

Methodology:

1) In the methodology section the "economic activity analysis" and the "occupation analysis" are too scarcely described. It is unclear to me why there is a need for both these analyses, and exactly what the differences are.

2) Moreover, there is not enough information on what the National Institute of Social Security contains of information: Gender? Age? Occupation? Socio-economic status?

3) The National Institute of Social Security is shortened to INSS at the start of the section, but throughout the rest of the paper the authors refer to something they call “social security”. Is this the same? If so, why not use INSS, or why not write National Institute of Social Security (hereafter referred to as “social security”)?

4) Finally, the bullet points make the paper look not ready for publication. This should rather be written in clear text, e.g., like: “The following rates of COVID-19 temporary disability processes were obtained as rates by (1) economic activity, and (2) occupation. Economic activity was defined as …”

Results:

1) The start of the results section with descriptive information works well.

2) However, the rest of the section feels too much of a series of listings, which can disrupt the flow of the text. I would rather suggest trying to get a better structure by grouping together similar occupations, or explicitly state the structure of the results. “Highest TD”, “Highest illness”, etc.

3) In Table 1 it is hard to understand what are decimals and what are not. I would suggest not using punctuation mark as a thousand separator, but rather using a space: 42 000 instead of 42.000.

Discussion:

1) I like the comparison to other studies and the possible explanations as to why some numbers are high. This enhances the quality of the paper.

2) “This analysis of TD reveals differences by gender, economic activity and occupation, allowing us to understand the work settings that have posed a higher risk of exposure and, indirectly, identify areas of COVID-19 transmission not adequately captured by other source of information (4).» Here it is pivotal to note what differences. Not just that the paper reveals differences.

3) On the other hand, I am not sure that the paper allows the authors to state this, as they have not adjusted for things like e.g., socio-economy. To illustrate this, in Norway during the first wave of the pandemic dentists were disproportionately affected on infections. People believed that this was due to the fact that they were so close to a patient’s mouth and nose and with a high risk of being infected. However, as more details emerged, it became apparent that dentists, often belonging to higher socio-economic brackets, had participated in skiing holidays in the Austrian Alps, which marked the onset of one of Europe's first super-spreader events.

4) “We also show that TD is a valuable supplementary source of information for analyzing the impact of COVID-19 in work settings different from health care sector” How?

5) The second paragraph feels just like a repetition of the results.

6) Temporary employment agency activities seem like a rather diffuse category. Can you include some examples: e.g.: “For example, Temporary employment agency activities, including occupations such as X, Y and Z, recorded a rate 9 times ….”

General remarks:

1) English should be written better and spelling errors must be checked. E.g. disability is written with one T, not two. Words like “objetive” instead of “objective”; and “previosly” instead of “previously”.

Reviewer #3: Introduction

1) In the introduction, in order to make the problem understandable and to introduce readers to the presented COVID absenteeism problem in Spain, some international (European) comparisons are useful.

2) The study sounds like a report in some places. In the introduction more motivations is useful, i.e. about the covid preliminary socio-economic consequecnes to Spanish society.

3) "...in quarantine and observation due to close contact with cases, and especially sensitive (vulnerable) workers..." What does "especially sensitive (vulnerable) workers" mean? please explain.

4) "In previous studies, different..." In order to understand and evaluate the research gap that this paper fills, more details about these studies are useful.

Methodology

1) "The ratio between TD due to quarantine and TD due to illness, defined as the ratio between TD

due to quarantine and TD due to illness in the last 7 days, was calculated. For every economic

sector of activity the median and interquartile range (IQR), representing the difference between

the third quartile (Q3) and the first quartile (Q1), were calculated. This allowed comparing the

dispersion among sectors.

Population figures affiliated with social security, disaggregated at 1, 2, and 3 NACE digits, were

used as denominators for rates calculations (15). To calculate rates by occupation, figures of

employed individuals at 1, 2, and 3 ISCO digits were used, provided by the Economically Active

Population Survey (16)."

In order to make the paper reader friendly (also for foreginers), these figures are useful. In addition, the equations for the mentioned ratios and statistical measures are necessary.

2)"The following rates of COVID-19 temporary..." equations needed.

Results

1) "Among the remaining cases, 74.4% (837,787) were due to

quarantine from close contact with COVID-19 cases, and 25.6% (288,969) were due to illness. Of

the TD processes, 54.6% (615,083) were observed in men (representing 54% of the studied

working population), and 45.4% (511,672) in women (46% of that working population). The TD

rate reached values of 5,465 quarantines and 1,878 illnesses per 100,000 women, and 4,883

quarantines and 1,690 illnesses per 100,000 men."

In the introduction, the authors mention three groups to be analyzed in the study, two of which are presented here (" and especially sensitive (vulnerable) workers." are omitted).

Discussion

1) Please separate discussion from conclusions.

2) Add some directions for future research

3) "The higher impact on the younger population

might be influenced by increased exposure, because from very early stages in the pandemic, it

was observed that the disease was more severe in older individuals, and they were likely

protected from exposure to some extent"

Are there any motivations in literature? Is this situation specific to Spanish society? Please explain.

4) "The results also show the uneven impact of COVID-19 by the economic activity of the company

or the occupation of the worker. The greatest impacts were observed in sectors such as Activities

of households as employers of domestic personnel, Temporary employment agency activities,

Waste Collection, Activities of call centres, and Processing and preserving of meat and meat

products, essential sectors that have not interrupted their on-site work at any point in the

evolution of the epidemic, either because telework is not possible (or more challenging), or

because the activity has been more intense due to the impact of the pandemic.

In this group of activities, it is worth noting the TD rate for domestic employees (NACE T and 97),

which is unusually high, especially in women, likely due to an overestimation resulting from not

including all workers in the sector in the denominator. It should be noted that affiliation data

does not include special regimes, which may be numerous for this group. On the other hand, we

have excluded and do not show the rates corresponding to Activities of extraterritorial

organizations (NACE U) and Armed Forces Occupations (ISCO Q), as they present extremely

small TD values, 99 and 39 processes, respectively, likely due to a notification bias.

Sectors such as Education or Food and beverage service activities show a lower impact. In the

case of Education, during the second wave, TD increased from late August and early September,

possibly indicating the massive screenings that many authorities carried out on teachers before

the start of the school year. In the case of Food and beverage service activities, it should be

considered that a large number of establishments were closed during both the first and second

waves, reducing the professional exposure of this group to COVID-19 and the incidence of TD"

These three paragraphs should be references in the literature.

6. PLOS authors have the option to publish the peer review history of their article (what does this mean?). If published, this will include your full peer review and any attached files.

Reviewer #1: **Yes: **Dr. med. Falk Liebers, MSc.

Reviewer #2: **Yes: **Fredrik Methi

Reviewer #3: No

---

## [Author Response · Author response to Decision Letter 0]

26 Jun 2024

Rebuttal letter

Luca Valera

Academic Editor

PLOS ONE

Dear Editor;

After having carefully reviewed each of the comments received, we have prepared a rebuttal letter, in which we explain each of the changes made to the suggestions of the editors and reviewers of the article. 

We are grateful for each of the comments, confident that they will contribute to improving the quality of information we intend to communicate.

Kind regards.

Dra. Montserrat García Gómez.

 Chief of the Occupational Health Area.

 Subdirectorate General of Environmental Health and Occupational Health.

 General Directorate of Public Health and Health Equity.

Ministry of Health.

 Paseo del Prado 18-20 

 ZIP 28014

MINISTERIO

DE SANIDAD

Revisión EDITOR

Comment 1:

Reply: We reviewed the suggested requirements and we believe that the new document meets the journal criteria.

Comment 2:

2. Please amend your manuscript to include your abstract after the title page.

Reply: We have included the abstract after the title page.

Comment 3:

3. In the online submission form you indicate that your data is not available for proprietary reasons and have provided a contact point for accessing this data. Please note that your current contact point is a co-author on this manuscript. According to our Data Policy, the contact point must not be an author on the manuscript and must be an institutional contact, ideally not an individual. Please revise your data statement to a non-author institutional point of contact, such as a data access or ethics committee, and send this to us via return email. Please also include contact information for the third party organization, and please include the full citation of where the data can be found.

Reply: Thank you for this comment. A statement will be included in the corresponding website section regarding data availability.

“The data underlying the results presented in the study are available, after request, from the social security website: 

https://sede.seg-social.gob.es/wps/portal/sede/sede/Ciudadanos/2022estadisticas/090316_c_ai?

changeLanguage=en” 

Comment 4:

Reply: We have changed in the text the captions for supplementary material to S1 Table and S2 Table. 

We have included supporting information at the end of manuscript

Reviewers' comments:

Reviewer's Responses to Questions

Comments to the Author

1. Is the manuscript technically sound, and do the data support the conclusions?

Reviewer #1: Yes

Reviewer #2: Partly

Reviewer #3: Partly

2. Has the statistical analysis been performed appropriately and rigorously?

Reviewer #1: Yes

Reviewer #2: No

Reviewer #3: No

3. Have the authors made all data underlying the findings in their manuscript fully available?

Reviewer #1: Yes

Reviewer #2: Yes

Reviewer #3: No

4. Is the manuscript presented in an intelligible fashion and written in standard English?

Reviewer #1: Yes

Reviewer #2: No

Reviewer #3: Yes

5. Review Comments to the Author

Reviewer #1: Peer-Review for PLOS ONE

Date of the review

• 31.01.2024

Title of the manuscript

• "Health impact of the COVID-19 in Spanish workers by gender: use of sickness absence for surveillance.”

Manuscript Identifier:

• PONE-D-24-01071

• First draft

Authors of the draft:

• Not blinded to the reviewer

Type of the manuscript

• primary study

Reviewer's report

Thank you for the opportunity to review the manuscript.

The aim of the study is describing the impact of the pandemic on the Spanish non-healthcare workers measured through health-related workplace absences based on disability data provided by the National Social Security. Temporary disability rates due to Covid-19 by economic activity and be occupation and ratios of the distribution of temporary disability due to quarantine and to Covid-19-illness are used to describe risks.

Scientific background, the aim of the study, hypothesis, statistical methods, results and the discussion are understandable, well and compact written and provide an impressive and good scientific view to the topic.

The reviewers have no additional considerations, recommendations or comments on the title, abstract, keywords, introduction, objective, methods, tables and the figure, the discussion and other aspects (references, acknowledgements etc.). There are no points left. The reviewer recommends the publication of the manuscript in PLOS ONE.

Reply: 

Thank you very much for your comments.

Reviewer #2:

 In this study the authors have investigated the share of temporary disability among non-healthcare working population in the two first waves of the pandemic in Spain. Temporary disability is defined as persons either in quarantine or sick due to covid-19 infection. The authors have investigated this based on gender, age, and occupation.

After carefully reading the paper, the general feeling is that the paper seems a bit unfished. There are several spelling errors, repetitive sentences, and the paper is missing a “story”. I therefore do not believe the paper is ready to be published yet.

Comment 1:

Introduction:

1) The introduction is working fine. However, I miss a specific knowledge gap this paper is trying to fill. Why is this important and needed? Why is there a need to look at both economic activity and occupation? What new and needed information does this bring?

Reply: Thank you for this comment and the relevant questions. We have adapted a paragraph with the suggested information in the introduction:

“In previous studies, different papers have described the impact of COVID-19 in terms of TD in the healthcare and sociosanitary population (7-12). However, it is important to highlight that the beginning of the second wave in Spain and later in Europe had its origin in Agriculture seasonal workers (6). This shows the relevance of monitoring the work settings and the working and living conditions of workers to better understand the COVID-19 transmission mechanisms. Notwithstanding this, there are very few studies describing the magnitude of the impact on the non-healthcare working population (6, 13, 14). Consequently, our current knowledge of the COVID-19 impact in these workers is still incomplete.”

Comment 2:

Methodology:

1) In the methodology section the "economic activity analysis" and the "occupation analysis" are too scarcely described. It is unclear to me why there is a need for both these analyses, and exactly what the differences are.

Reply: 

Thank you for your comments, we have reorganised the second and third paragraph of the “Methodology” section in order to make it easier to understand: 

 “We use the European Classification of Economic Activities (NACE) (13) for the analysis by economic activity and the International Classification of Occupations (ISCO) (14) for the analysis by occupation. These classifications provide good indicators of the working conditions of workers in a complementary way. 

For the economic activity analysis, healthcare and sociosanitary activities were excluded (Q86, Q87, and Q88 codes from the NACE. For the occupation analysis, healthcare and personal care professionals were also excluded (codes 21, 56 and 57 of the ISCO).”

Comment 3:

2) Moreover, there is not enough information on what the National Institute of Social Security contains of information: Gender? Age? Occupation? Socio-economic status?

Reply: 

The social security database contains information structured in multiple variables. For achieving the objectives of this study, we requested only several needed variables: age, gender, NACE code, ISCO code and ICD code (to determine the sickness absence nature). We have included the following information in the first paragraph of the methodology, in order to clarify this point:

“The cases under study were the processes of temporary disabillity due to COVID-19 (ICD-10 ES codes: B34.2, B97.21, and U07.1) and quarantine due to close contact with a COVID-19 case (ICD-10 ES code: Z20.828) from February 15 to September 17, 2020, provided by the National Institute of Social Security (INSS), by age, gender, economic activity and occupation.” 

Comment 4:

3) The National Institute of Social Security is shortened to INSS at the start of the section, but throughout the rest of the paper the authors refer to something they call “social security”. Is this the same? If so, why not use INSS, or why not write National Institute of Social Security (hereafter referred to as “social security”)?

Reply: 

Thank you for this coment. We checked the text and used the INSS term when we refer to the public institution and “social security” when we mean the service that the institution offers.

Comment 5:

4) Finally, the bullet points make the paper look not ready for publication. This should rather be written in clear text, e.g., like: “The following rates of COVID-19 temporary disability processes were obtained as rates by (1) economic activity, and (2) occupation. Economic activity was defined as …”

Reply: We modified the text in order to eliminate the bullet points:

“The following rates of COVID-19 temporary disability processes were obtained as rates by (1) economic activity, and (2) occupation. 

COVID-19 Temporary disabillity Rate by Economic Activity (TDREA) was defined as: Number of temporary disabillity processes (NTD) in a NACE code per 100,000 active affiliates to Social Security in that NACE code. 

TDREA=(NTD by NACE code*100,000)/(Number of active affiliates to Social Security by NACE code )

COVID-19 Temporary disabillity Rate by Occupation (TDRO) was defined as: Number of temporary disabillity processes in an ISCO code per 100,000 employed individuals in that ISCO code.

TDRO=(NTD By ISCO code*100,000)/(Number of employed individuals by ISCO code )

“

Comment 6:

 Results:

1) The start of the results section with descriptive information works well.

2) However, the rest of the section feels too much of a series of listings, which can disrupt the flow of the text. I would rather suggest trying to get a better structure by grouping together similar occupations, or explicitly state the structure of the results. “Highest TD”, “Highest illness”, etc.

Reply: We believe that you are right to look for better groupings and that is what we have tried to do throughout the writing of the article, however, there is a lot of heterogeneity in the results so that the information we present is already a summary of a large number of occupational sectors. The occupations presented already constitute a grouping according to occupation categories.

Comment 7:

3) In Table 1 it is hard to understand what are decimals and what are not. I would suggest not using punctuation mark as a thousand separator, but rather using a space: 42 000 instead of 42.000.

Reply: 

Thank you for this observation. We have standardised the criteria according to the magazine's suggestions.

Comment 8:

Discussion:

1) I like the comparison to other studies and the possible explanations as to why some numbers are high. This enhances the quality of the paper.

2) “This analysis of TD reveals differences by gender, economic activity and occupation, allowing us to understand the work settings that have posed a higher risk of exposure and, indirectly, identify areas of COVID-19 transmission not adequately captured by other source of information (4).» Here it is pivotal to note what differences. Not just that the paper reveals differences.

Reply: Thank you for this coment. We have added additional information in order to point out these important differences in the first paragraph of the Discusion:

“The results revealed the high impact in terms of sick leave associated with COVID-19 in Spanish workers and important differences by gender, economic activity and occupation that have characterized this impact. The TD rate was 26 and 9.5 times higher for women than for men in the Activities of households as employers of domestic personnel and in the Temporary employment agency activities, respectively. Waste Collection, Activities of call centers, and Processing and preserving of meat products, were also sectors with high risk of exposure for both genders. Taking into account the workers occupation, the less qualified presented the highest rates.”

Afterwards, we further explore and discuss these differences in the rest of the discution section. 

Comment 9:

3) On the other hand, I am not sure that the paper allows the authors to state this, as they have not adjusted for things like e.g., socio-economy. To illustrate this, in Norway during the first wave of the pandemic dentists were disproportionately affected on infections. People believed that this was due to the fact that they were so close to a patient’s mouth and nose and with a high risk of being infected. However, as more details emerged, it became apparent that dentists, often belonging to higher socio-economic brackets, had participated in skiing holidays in the Austrian Alps, which marked the onset of one of Europe's first super-spreader events.

Reply: We agree with your statement and are aware that the events you mention are real and may influence the analysis of the results. We have included a comment within the limitations that relates to your comments: 

“Associating infections with places and jobs, that is, with certain working conditions, does not incorporate living conditions, commutes to workplaces, and other socioeconomic and environmental determinants, which are relevant to characterize the pandemic and were relevant in outbreaks of slaughterhouses and seasonal workers, for example.”

Comment 10:

4) “We also show that TD is a valuable supplementary source of information for analyzing the impact of COVID-19 in work settings different from health care sector

---

## [Editor Report · Decision Letter 1]

2 Jul 2024

Health impact of the COVID-19 in Spanish non-healthcare workers by gender: use of sickness absence for surveillance.

PONE-D-24-01071R1

Dear Dr. Culqui,

We’re pleased to inform you that your manuscript has been judged scientifically suitable for publication and will be formally accepted for publication once it meets all outstanding technical requirements.

Kind regards,

Luca Valera

Academic Editor

PLOS ONE

Additional Editor Comments (optional):

Thank your for addressing all the reviewers' suggestions and commentaries. The paper should be published as it is.
---

## [Editor Report · Acceptance letter]

7 Aug 2024

PONE-D-24-01071R1 

PLOS ONE

Dear Dr. Culqui L., 

I'm pleased to inform you that your manuscript has been deemed suitable for publication in PLOS ONE. Congratulations! Your manuscript is now being handed over to our production team.

Kind regards, 

on behalf of

Dr. Luca Valera 

Academic Editor

PLOS ONE